# A Subpathway and Target Gene Cluster-Based Approach Uncovers lncRNAs Associated with Human Primordial Follicle Activation

**DOI:** 10.3390/ijms241310525

**Published:** 2023-06-23

**Authors:** Li Zhang, Jiyuan Zou, Zhihao Wang, Lin Li

**Affiliations:** Guangdong Provincial Key Laboratory of Proteomics, Department of Pathophysiology, School of Basic Medical Sciences, Southern Medical University, Guangzhou 510515, China; bio_minizhang@outlook.com (L.Z.); zouzouzoujiyuan@163.com (J.Z.); vinod_wang@163.com (Z.W.)

**Keywords:** lncRNA, subpathway, target gene cluster, primordial follicle activation, premature ovarian insufficiency

## Abstract

Long non-coding RNAs (lncRNAs) are emerging as a critical regulator in controlling the expression level of genes involved in cell differentiation and development. Primordial follicle activation (PFA) is the first step for follicle maturation, and excessive PFA results in premature ovarian insufficiency (POI). However, the correlation between lncRNA and cell differentiation was largely unknown, especially during PFA. In this study, we observed the expression level of lncRNA was more specific than protein-coding genes in both follicles and granulosa cells, suggesting lncRNA might play a crucial role in follicle development. Hence, a systematical framework was needed to infer the functions of lncRNAs during PFA. Additionally, an increasing number of studies indicate that the subpathway is more precise in reflecting biological processes than the entire pathway. Given the complex expression patterns of lncRNA target genes, target genes were further clustered based on their expression similarity and classification performance to reveal the activated/inhibited gene modules, which intuitively illustrated the diversity of lncRNA regulation. Moreover, the knockdown of *SBF2*-*AS1* in the A549 cell line and *ZFAS1* in the SK-Hep1 cell line further validated the function of *SBF2*-*AS1* in regulating the Hippo signaling subpathway and *ZFAS1* in the cell cycle subpathway. Overall, our findings demonstrated the importance of subpathway analysis in uncovering the functions of lncRNAs during PFA, and paved new avenues for future lncRNA-associated research.

## 1. Introduction

Follicle maturation refers to the development of primordial follicles into preovulatory follicles, where follicles and granulosa cells undergo extensive gene expression changes [1,2,3]. Primordial follicles form around birth and are non-renewable. The majority of primordial follicles maintain a dormant state. Primordial follicle activation refers to the process by which primordial follicles change from a quiescent to a growing state, i.e., the transition from a primordial follicle to a primary follicle. Primordial follicle activation (PFA) is a prerequisite for folliculogenesis, which refers to the process where follicle size enlarges, and the morphology of granulosa cells becomes cuboidal [4]. Researchers have indicated that excessive activation of the primordial follicles resulted in premature ovarian insufficiency (POI) and, ultimately, female sterility [5]. Studies indicate that POI patients still have some residual primordial follicles in their ovaries, which provides a new direction for the treatment by promoting the activation of these primordial follicles [6,7]. Therefore, understanding the mechanisms driving PFA is of particular importance.

Long non-coding RNAs (lncRNAs) are operationally defined as non-coding RNA transcripts longer than 200 nucleotides (nt). LncRNAs are involved in cell commitment by regulating the expression of certain genes [8,9]. Increasing studies have indicated that lncRNAs showed higher lineage specificity in multiple human organs than protein-coding genes [10]. However, limited reports focus on lncRNA during PFA. Therefore, there is an urgent need to develop computational approaches to explore the function of lncRNA during PFA.

LncRNA exerts its function by regulating hundreds or even thousands of target genes, while these target genes may have opposite expression patterns under certain biological contexts [11,12]. Therefore, it is necessary to divide target genes into different groups based on their expression value.

Pathway analysis is widely used to interpret relevant biological changes under certain contexts [13,14]. However, the entire pathway is often too large to accurately elucidate relevant biological regulation. A local region with specific biological functions is defined as a subpathway and may be more suitable for uncovering relevant biological events. Recently, a growing body of research indicated that subpathway analysis was more effective in elucidating the pathogenesis of disease and identifying cancer risk gene markers [15,16,17]. However, the application of subpathways to elucidate the mechanism of reproductive cell development is still lacking. Moreover, there were limited subpathway analysis studies at the single-cell level.

Collectively, to further improve our understanding of lncRNAs involved in stepwise and follicle maturation, a subpathway and target gene cluster-based approach was applied to predict lncRNA–subpathway connections, which provides new insight into the underlying mechanisms of lncRNAs during PFA more precisely.

## 2. Results

### 2.1. LncRNAs Improve Cell Type Classification during Human Follicle Maturation

We obtained follicles and corresponding granulosa cells from the primordial to preovulatory stages in GSE107746 [1]. In order to reveal whether lncRNA is cell-specific during follicle maturation, we performed UMAP analysis to visualize the distribution of follicles and granulosa cells from the primordial to preovulatory stages based on protein-coding genes and lncRNAs. We observed that the lncRNA signature displayed higher cell type specificity than protein-coding genes (Figure 1A). The standard deviation (sd) value of lncRNAs was lower than that of protein-coding genes, indicating that the expression pattern of lncRNA was more consistent in cells (Figure 1B). The number of expressed lncRNAs gradually increased in follicles from the primordial to preovulatory stages (see the method in Figure 1C). In order to rule out the effects of follicular size on the number of expressed protein-coding genes and lncRNAs, the attribute of nFeature_RNA of was calculated, and we found that the number of expressed lncRNAs was independent of follicular size (Appendix A). In addition, there were fewer overlapping lncRNAs between the follicles and corresponding granulosa cells from the primordial to preovulatory stages (Figure 1D). Collectively, lncRNA showed higher cell type specificity than protein-coding genes during follicle maturation.

### 2.2. Distinct Transcriptomic Expression of lncRNAs during PFA

PFA is a crucial step for female fertility [18], and excessive PFA results in POI. Increasing evidence indicates that PFA is a tightly controlled process [19,20]. However, the exact mechanism of controlling the selective activation of primordial follicles mediated by lncRNAs is unknown. According to Appendix A, from primordial to primary stages, unions of 648 expressed lncRNAs in follicles and 170 lncRNAs in granulosa cells were acquired. By comparing the expression level of lncRNA between follicles and granulosa cells, we identified 65 upregulated lncRNAs in follicles and 22 upregulated lncRNAs in granulosa cells during PFA (Figure 2A, Appendix A). For example, *SBF2-AS1*, *MRVI1*-*AS1*, *CASC8*, *TEX41*, *CYP1B1*-*AS1*, and *OIP5*-*AS1* were mainly expressed in follicles, whereas *WT1*-*AS*, *THAP9*-*AS1*, *GAS5*, *MALAT1*, *ZFAS1*, *MIR22HG,* and *SNHG8* had higher expression level in granulosa cells. Among them, *CASC8* was specifically expressed in MII oocytes [21]. Moreover, across nine GTEx tissues (heart, liver, spleen, lung, kidney, testis, vagina, uterus, and ovary), we noticed that the majority of these lncRNAs were expressed in ovary tissues (Figure 2B). Hence, the expression of these lncRNAs at the single-cell and bulk levels provided a prelude to indicate the importance of lncRNA during PFA.

### 2.3. Identifying Primordial Follicle Activation Associated lncRNA Clusters

LncRNAs play crucial roles in gene regulation at post-transcriptional levels [22,23]. In this study, bioinformatics analysis used the competitive endogenous RNA (ceRNA) theory to infer the lncRNA target genes. During PFA, a total of 97 lncRNA-associated target gene pairs in follicles and 47 lncRNA-corresponding target gene pairs in granulosa cells were identified (see details in Section 4). Given the expression diversity of target genes, we applied a four-step computational framework that divided lncRNA target genes into different clusters based on the genes’ PCC similarity and classification performance (AUC value) during PFA (Figure 3A). The number of lncRNA target gene clusters varied from 1 to 6 in follicles and 1 to 11 in granulosa cells during PFA (Figure 3B). LncRNA target genes for each cluster were indicated. For example, *SBF2*-*AS1* was highly expressed in primordial follicles and primary follicles. There were five target gene clusters of *SBF2*-*AS1*. Among these clusters, *PTGS2* in cluster 1 played a crucial role in the acquisition of oocyte competence [10]. *IGF1R* and *KIT* in cluster 2 were fertility-associated genes. In addition, *ZFAS1* was mainly expressed in granulosa cells with three target gene clusters (Appendix A). Target genes in cluster 1 were mainly activated in primary granulosa cells, and genes like *BECN1*, *CDKN1B*, *EIF2B2*, and *MTRR* were POI-related genes [24]. These results suggested that grouping lncRNA target genes into different clusters was beneficial for further analysis of follicle maturation-associated functional gene modules.

### 2.4. Differences between Subpathway and Pathway during Follicle Maturation

Pathway identification may improve our understanding of lncRNA function. However, the traditional biological pathway is complex and covers multiple genes. A growing number of studies have indicated that the cell fate transition was often caused by changes in local regions (subpathways) within the biological pathway rather than the whole pathway [25,26]. However, the detailed mechanism of subpathway activation required further investigation.

In this study, 725 subpathways were identified from 238 pathways by the R package psSubpathway (Figure 4A). Each pathway contained 1–20 subpathways, which consisted of 3 to 399 genes. Then, a total of 210 subpathways in follicles and 234 subpathways in granulosa cells were significantly activated during PFA ((Formula (2) in Materials and Methods, *p* < 0.05). Studies have reported that the “cell cycle pathway”, “mTOR signaling pathway”, “ErbB signaling pathway”, and “FoxO signaling pathway” were associated with follicle growth [1,27,28,29,30]. Comparing the ssGSEA (single sample Gene Set Enrichment Analysis) score and pathway specificity score between pathway and corresponding subpathways, we observed high heterogeneity of activation among the pathway and corresponding subpathways in follicles, which was low in granulosa cells (Figure 4B and Appendix A). We noticed that the expression level of ligand/receptor was relatively conserved in granulosa cells in Zhang et al.’s study [1], and the UMAP distribution of granulosa cells was adjacent from the primordial to preovulatory stages using protein-coding genes (Figure 1A), suggesting that the differentiation of granulosa cells did not change much. Hence, we hypothesized that the activation pattern of the subpathway in granulosa cells showed lower heterogeneity, suggesting that the transcriptome changes in granulosa cells were relatively stable in inducing follicle maturation. In summary, these results indicated that it is more precise to depict the follicle fate transition using subpathways.

### 2.5. The lncRNA–Subpathway Network and Its Characteristics during Primordial Follicle Activation

After identifying the subpathways dynamically activated during PFA, we investigated whether the expressed lncRNA-target gene clusters work together to regulate a specific subpathway. In this study, a hypergeometric test was used to measure the statistical significance of shared protein-coding genes between subpathway genes and lncRNA target cluster genes. Then, 91 lncRNAs involved in 210 dynamic subpathways were obtained in follicles, and 17 lncRNAs participating in 234 dynamic subpathways were identified in granulosa cells. We observed subpathways associated with MAPK, Insulin, Ras, Autophagy, Hippo, Cell cycle, and mTOR were activated in follicles and granulosa cells during PFA (Figure 5A). When further ranking these lncRNAs by Formula 5 based on the hypergeometric *p*-value and the corresponding number of dynamic subpathways, the top 20 lncRNAs (20/91) in follicles (Appendix A) and 17 lncRNAs in granulosa cells are shown in Figure 5B. The overlapping genes of lncRNA target genes with subpathway genes and POI genes are illustrated (Figure 5C). We found lncRNAs regulated genes involved in the POI phenomenon, reflecting how these lncRNAs might play an important role in female fertility. Among these POI-related genes in Figure 5C, *CDKN1B*, *FMR1*, *FOXO3*, *NOBOX,* and *LARS2* was candidate pathogenesis for POI [31,32,33,34,35]. Moreover, we observed the target genes of *CASC8* were associated with “Thermogenesis” by regulating *ACTL6A*. *FGD5*-*AS1* was involved in the “MAPK_signaling_pathway (04010_21)” by regulating MAPK genes like *CRKL*, *HSPA8*, *MAPK8*, *MAX*, and *NFATC3*. *FGD5*-*AS1* affected the expression of POI-related genes (*EIF2B2*, *EXO1*, *FMR1*, and *FOXO3*). These findings provided detailed lncRNA–subpathway connections during PFA.

### 2.6. Validating the Function of lncRNAs Based on Subpathway Genes after lncRNA Knockdown

In our study, *SBF2*-*AS1* was associated with the “Hippo signaling pathway (04390_1)” and “Regulation of actin cytoskeleton (04810_13)”. *ZFAS1* was associated with the “mTOR signaling subpathway (04150_16)” and “Cell cycle (04110_3)” (Figure 5C). In Figure 2B, we can observe that *SBF2*-*AS1* and *ZFAS1* were not only expressed in ovarian tissues but also in liver and lung tissues. Through comprehensively searching and reviewing currently available lncRNA knockdown datasets, we obtained SBF2-AS1 knockdown data in the human A549 cell line (GSE103016) and *ZFAS1* knockdown data in the human SK-Hep1 cell line (GSE104226) [36]. To further validate the potential function of *SBF2*-*AS1* and ZFAS1, we explored the expression level of subpathway genes after the lncRNA knockdown (Figure 6). Consistently, we noticed that the knockdown of *SBF2*-*AS1* and *ZFAS1* induced down-regulation of corresponding subpathway genes. For example, *DLG3*, *FBXW11*, *BTRC*, *PPP2R2A*, *RASSF6,* and *YAP1* belonged to the Hippo signaling pathway (04390_1). These genes were down-regulated after *SBF2*-*AS1* knockdown, among which *PPP2R2A* and *RASSF6* were target genes of *SBF2*-*AS1* during PFA. These results confirmed the inferred connection between lncRNAs and subpathways.

## 3. Discussion

LncRNAs are non-coding transcripts that have been studied extensively in the process of cell development [37]. Human follicles were divided into five stages based on follicular size and granulosa cell numbers, i.e., primordial follicles, primary follicles, secondary follicles, antral follicles, and preovulatory follicles [38,39]. PFA is necessary for follicle growth. However, systematic analysis of lncRNAs involved in PFA is still poorly understood. In this study, a comprehensive analysis of the expression level of lncRNAs and protein-coding genes in follicles and granulosa cells, we noticed that lncRNAs were more cell type-specific than protein-coding genes. Additionally, a growing number of studies have investigated the biological changes based on subpathways, which provided a clearer and more defined window to understand the biological process of follicle maturation. To decode the lncRNA mechanism during PFA, we obtained subpathways and further grouped lncRNA target genes into different clusters, where each gene cluster exhibited consistent expression patterns and high classification ability during PFA. The connection between subpathway genes and lncRNA was inferred based on enrichment analysis during PFA.

The development of follicle maturation is associated with multi-factorial and complex steps. Numerous crucial lncRNAs that modulated cell proliferation and differentiation have been identified. For example, *SBF2*-*AS1* and *MRVI1*-*AS1* were upregulated and expressed in follicles, whereas *WT1*-*AS* and *ZFAS1* were highly expressed in granulosa cells during PFA (Figure 2A). Chen et al. found that the silencing of *SBF2*-*AS1* reduced the proliferative ability of esophageal squamous cell carcinoma (ESCC) [40]. Tuo et al. observed that *MRVI1*-*AS1* promoted the progression of hepatocellular carcinoma (HCC) [41]. Lou et al. reported that the G1-S phase transition increased after the knockdown of *GAS5* in prostate cancer [42]. O’Brien indicated that the knockdown of *ZFAS1* was associated with decreased cellular proliferation [43]. However, more studies are still required to ascertain lncRNA function during PFA. In this study, we investigated the development of follicles and granulosa cells by performing UMAP analysis and found that lncRNAs could more effectively discriminate cell type than protein-coding genes. We revealed the dynamic expression of lncRNAs in follicles and granulosa cells during PFA (Figure 2A), suggesting the complexity of lncRNAs in related biological processes. For example, we observed that *SBF2*-*AS1*, *MRVI1*-*AS1*, and *CASC8* were highly expressed in follicles, whereas *WT1*-*AS*, *THAP9*-*AS1*, and *GAS5* were upregulated in granulosa cells. *SBF2*-*AS1* was found to be associated with cell proliferation and epithelial-to-mesenchymal transition (EMT) [44,45,46]. Wang et al. found the overexpression of *GAS5* contributed to increased expression levels of IL-6 in human granulosa cells (KGN) [42]. Moreover, we divided lncRNA target genes into different clusters based on genes’ PCC similarity and classification performance. Distinct expression patterns of genes in each cluster suggested the functional diversity of lncRNAs in regulating gene expression.

Additionally, a body of studies reported that subpathways were more suitable for interpreting underlying biological processes. In this study, we obtained subpathways and first visualized the heterogeneity of activation patterns among pathways and subpathways using the ssGSEA score and specificity score. We noticed that the activation pattern of the subpathway in follicles showed higher heterogeneity than that in granulosa cells. Although subpathway activation had individual differences, some subpathways associated with cell cycle, mTOR, and Hippo were co-activated between granulosa cells and follicles during PFA. The predicted connection between lncRNAs and subpathways was further validated by lncRNA knockdown cell lines.

Currently, the rate of infertility is high all over the world. Oocyte maturation failure is a contributing factor to female infertility [47]. In vitro maturation (IVM) and ovarian stimulation technology have been proposed to treat oocyte maturation failure by activating dormant ovarian follicles to generate mature eggs for successful reproduction [48,49,50,51]. In this study, we found lncRNAs played an important role in PFA, offering a new avenue for inducing follicle maturation.

In summary, based on subpathway specificity and target gene expression consistency, we built a systematic approach to uncover lncRNA functions during PAF (Figure 7). Our subpathway analysis seemed to be the first implementation associated with follicle maturation. In parallel, the pipeline is also suitable for analyzing lncRNA functions in other patterns of cell fate determination.

## 4. Materials and Methods

### 4.1. Data Collection and Pre-Processing

The expression dataset of human ovarian follicles and matched granulosa cells from primordial to preovulatory stages at single cell level was downloaded from the Gene Expression Omnibus (GEO) database (https://www.ncbi.nlm.nih.gov/geo/, accessed on 15 September 2022) under accession number GSE107746 [1], including primordial follicles (n = 17), primary follicles (n = 25), secondary follicles (n = 12), antral follicles (n = 23), preovulatory follicles (n = 3), primordial granulosa cells (n = 8), primary granulosa cells (n = 15), secondary granulosa cells (n = 6), antral granulosa cells (n = 24), and preovulatory granulosa cells (n = 18). In this study, 18,797 protein-coding genes and 13,870 lncRNAs were annotated in GTF file downloaded from GENCODE database (https://www.gencodegenes.org/, v19, accessed on 10 March 2023). Fragments per kilobase of transcript per million mapped reads (FPKM) values of protein-coding genes and lncRNAs were normalized by log2(FPKM + 1).

The raw count datasets of 9 human tissues (heart, liver, spleen, lung, kidney, testis, vagina, uterus, and ovary) were downloaded from the Genotype-Tissue Expression (GTEx) database (v8, accessed on 29 March 2022) [52]. The read count datasets were normalized by log2(Count + 1).

### 4.2. Cell Clustering Analysis

R package “Seurat” v3.2.1 was used to analyze the transcriptome data of human ovarian follicles and matched granulosa cells [53]. UMAP (Uniform Manifold Approximation and Projection) was applied to reduce the dimensionality. We calculated the nFeature_RNA and standard deviation (sd) attributes in each cell based on the expression of protein-coding genes and lncRNAs, respectively.

### 4.3. Identifying Follicle Maturation Associated Protein-Coding Genes and lncRNAs

To identify protein-coding genes and lncRNAs associated with follicle maturation, two filtering criteria were applied: (1) protein-coding genes/lncRNAs with an average expression value ≥ 1 in any cell type and (2) expressed in ≥70% cells of any follicle or granulosa cell type. Then, from primordial to preovulatory stages, we identified unions of 7730 protein-coding genes and 1497 lncRNAs in follicles, unions of 6522 protein-coding genes, and 729 lncRNAs in granulosa cells. Function ‘FindMarkers’ in Seurat with “wilcox” test differential expression analysis method was used to obtain differentially expressed lncRNAs (Bonferroni adjusted *p* < 0.05).

### 4.4. Identification of lncRNA Target Gene Cluster

The 743,029 miRNA–mRNA and 31,344 miRNA–lncRNA interaction pairs in our study were obtained from miRNet database (https://www.mirnet.ca/, accessed on 21 March 2023) [54]. Then, to explore competitive interaction between lncRNA and mRNA, hypergeometric test was used to measure the significance of shared miRNAs between lncRNA and mRNA as follows:(1)$$p=1-\sum_{i=1}^{r-1}\frac{\left(\begin{array}{c} t\\ i \end{array}\right)\left(\begin{array}{c} m-t\\ n-i \end{array}\right)}{\left(\begin{array}{c} m\\ n \end{array}\right)}$$
where *m* represents the total number of miRNAs in miRBase v22; *t* represents the number of miRNAs interacting with mRNAs; *n* represents the number of miRNAs interacting with the lncRNAs; and *r* represents the number of common miRNAs simultaneously shared by mRNAs and lncRNAs. We considered *p*-value < 0.05 to be statistically significant.

Given that lncRNA could regulate multiple genes, we divided lncRNA target genes into different clusters, as described in a previous study [55]. (1) Correlation analysis of related target genes was calculated using Pearson’s correlation coefficients (PCCs) in primordial and primary stages; (2) the hierarchical clustering was applied, where genes (≥3) with correlation coefficient above 0.7 were stratified into one group, while those cannot be grouped into any clusters were discarded; (3) the LIBSVM toolbox implemented in R package ‘e1071’ was adopted to perform the support vector machines (SVMs) classification to evaluate the classification performance of each gene clusters between primordial and primary stages; and (4) 5-fold cross validation with 100 repetitions were used to evaluate the classification performance. Subsequently, the mean of the 100 AUC scores was regarded as its final score. Target gene clusters of each lncRNA with average AUC scores larger than 0.8 were retained.

### 4.5. Obtaining Subpathway Based on k-Clique Algorithm

In this study, subpathways were extracted by R package psSubpathway [56]. In total, 725 subpathways were annotated from 238 pathways (Appendix A).

### 4.6. Subpathways Activity Score

To identify subpathways with dynamic changes during primordial follicle activation, the activity score for subpathways was defined as follows:(2)$$\mathrm{Acitivity}=\sum_{i=1}^{n}\frac{t_{i}}{n}$$
where *n* represents the number of genes in each subpathway, $t_{i}$ denotes the *t*-score of the *i*th gene in the subpathway calculated by Student’s *t*-test analysis for comparing the gene expression profiles between primordial stage and primary stage. Next, random selection of the same number of genes for each candidate pathway with 1000 repetitions was performed to calculate *p*-value of subpathway. In this study, 214 subpathways in follicles and 295 subpathways in granulosa cells (*p*-value < 0.05) were associated with PFA.

### 4.7. Cell Type-Specific Pathway Specificity

The pathway specificity score in each cell type was defined as follows:(3)$${\mathrm{Specificity}}_{ij}=\frac{E_{ij}}{E_{i}},\;k\in 1\;\ldots N\;$$
where $E_{ij}$ represents the average expression level of subpathway *i* genes in cell type *j*, and $E_{ik}$ represents the average expression level of subpathway *i* genes in all cells.

### 4.8. LncRNA Rank

After evaluating the importance of lncRNAs during primordial follicle activation, we ranked the lncRNAs by lncRNA rank (*LR*) score to evaluate their associations with dynamic subpathways. For each lncRNA, the score *LR* was defined as follows:(4)$$Entropy_{i}=-log10\left(q_{i}\right)$$
(5)$$LR=\frac{\sum_{i=1}^{n}Entropy_{i}\times P_{i}}{n}$$
where $q_{i}$ denotes the *p*-value of overlapping mRNAs between subpathway *i* genes and the lncRNA target gene clusters calculated by hypergeometric test; $P_{i}$ denotes the activity score of subpathway *i* in Formula (2), and *n* represents the number of all subpathways regulated by this lncRNA.

### 4.9. Premature Ovarian Insufficiency (POI)-Related Geneset

A total of 127 POI–related genes (Appendix A) in Zhao et al.’s study, which summarized genes from databases of NCBI Gene, OMIM, and DrugBank, were used [24].

### 4.10. Download lncRNA Knockdown Datasets

To further validate the potential function of *SBF2*-*AS1* and *ZFAS1* identified in this study, RNA-sequencing data analysis was conducted to infer the dynamic expression of subpathway genes after knockdown of *SBF2*-*AS1* in human A549 cell line (GSE103016) and *ZFAS1* in human SK-Hep1 cell line (GSE104226) [36].

### 4.11. Statistical Analysis

R software (www.rproject.org, version 3.6.3, accessed on 28 October 2020) was used for all statistical analyses. Wilcox test was used to identify differentially expressed lncRNAs. Student’s t-test was performed to uncover dynamic subpathways during PFA. Hypergeometric test was employed to detect the significance of shared miRNAs between lncRNAs and mRNAs, and shared mRNAs between subpathway genes and lncRNA target genes. *p*-value < 0.05 was used to indicate statistical significance in this study.

## Figures and Tables

**Figure 1 ijms-24-10525-f001:**
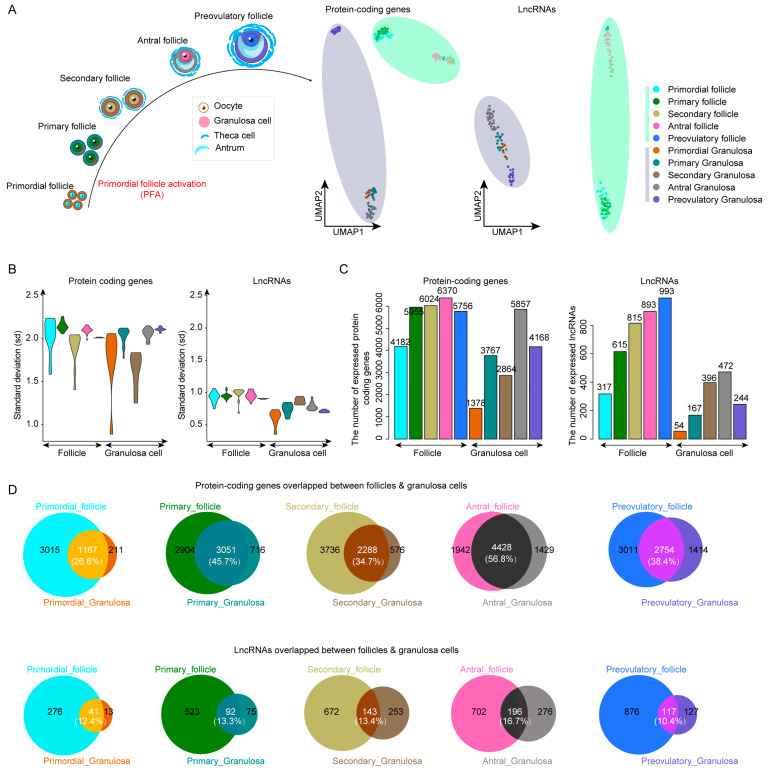
The specificity of lncRNAs during human follicle maturation. (**A**) The left panel shows the development of human follicle maturation. The right panel shows the UMAP plot of follicles and granulosa cells by protein-coding genes or lncRNAs. (**B**) Standard deviation analysis showing the distribution differences in each cell type by protein-coding genes and lncRNAs. (**C**) The number of expressed protein-coding genes and lncRNAs. (**D**) Upper Venn diagrams show the overlapped protein-coding genes between follicles and granulosa cells, and bottom Venn diagrams show the overlapping lncRNAs. From primordial to preovulatory stages, the turquoise1, dark green, darkkhaki, hotpink and royalblue1 colors indicate follicles. The strong orange, darkcyan, burlywood4, snow4 and slateblue colors indicate granulosa cells.

**Figure 2 ijms-24-10525-f002:**
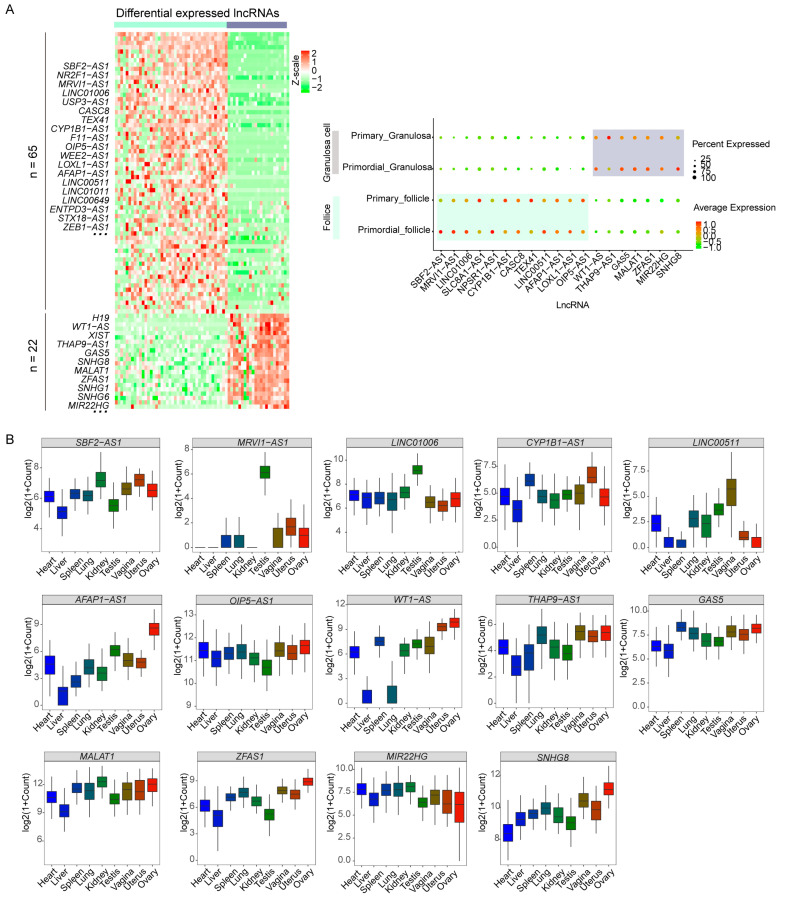
Dynamic expression of lncRNAs in follicles and granulosa cells during PFA. (**A**) The left heat map shows the typical differentially expressed lncRNAs (Wilcox test, Bonferroni adjust *p* < 0.05, see Appendix A). Red and green represent an increase and the decrease in the lncRNA expression levels, respectively. The bar on the top indicates cell types using lime green (follicles) and grey (granulosa cells). Right dot plot shows the expression pattern in follicles and granulosa cells during PFA. (**B**) The expression level of lncRNAs across 9 tissues in GTEx project.

**Figure 3 ijms-24-10525-f003:**
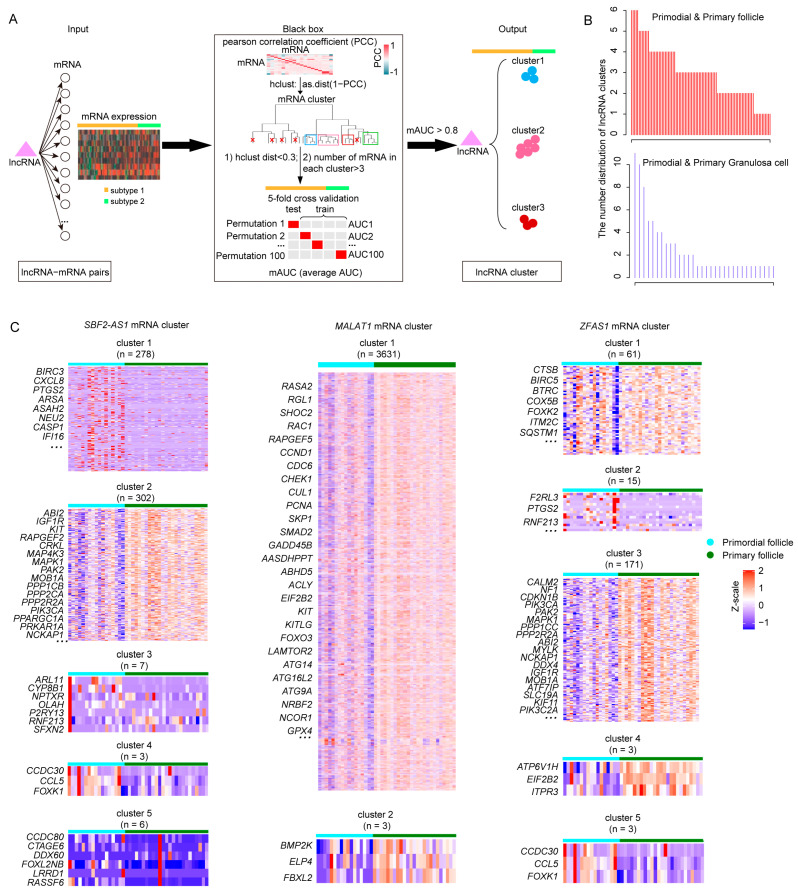
Pipeline of lncRNA target gene clusters. (**A**) Workflow of lncRNA target gene cluster identification. (**B**) The number distribution of lncRNA target gene clusters in follicles (upper panel) and granulosa cells (lower panel) during primordial follicle activation. (**C**) Heat maps of cluster genes of *SBF2*-*AS1*, *MALAT1*, and *ZFAS1* in primordial and primary follicles.

**Figure 4 ijms-24-10525-f004:**
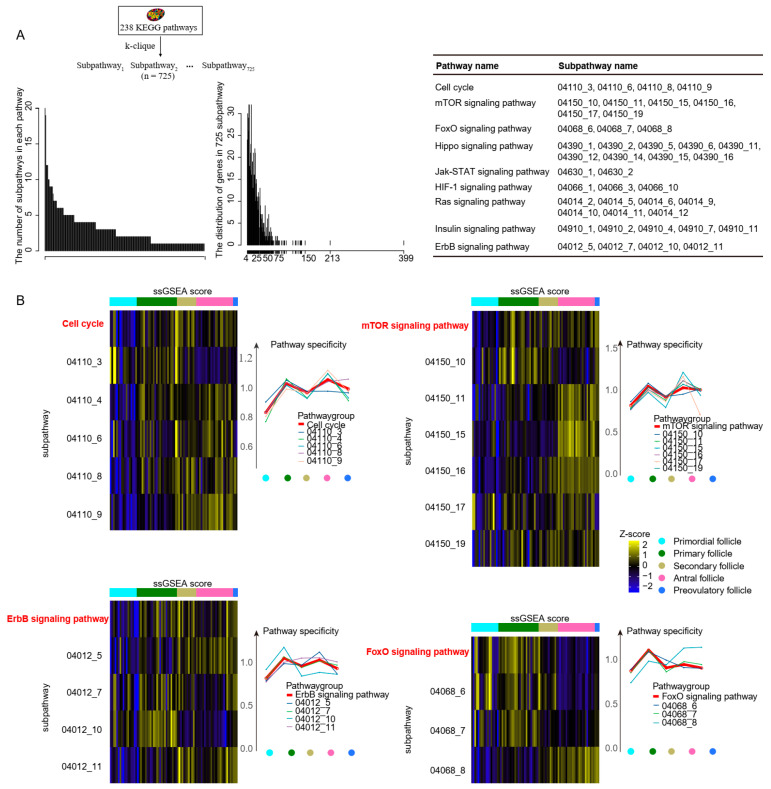
The dynamics of subpathways in follicles from primordial to preovulatory stages. (**A**) Identification of subpathway by k-clique algorithm. The left histogram is the distribution of the number of genes among 725 subpathways ranging from 3 to 399. The middle histogram is the number of subpathways among each pathway ranging from 1 to 20. The right table lists corresponding subpathways for a given pathway. (**B**) The difference between “Cell cycle”, “mTOR signaling pathway”, “ErbB signaling pathway”, and “FoxO signaling pathway” and their corresponding subpathways in follicles from primordial to preovulatory stages. Take “Cell cycle” pathway as example; the left heat map shows the ssGSEA score between “Cell cycle” pathway and 5 subpathways (04110_3, 04110_4, 04110_6, 04110_8, and 04110_9). The right line chart indicates the dynamic change in pathway specificity; red line represents the entire pathway, whereas other color lines represent corresponding subpathways.

**Figure 5 ijms-24-10525-f005:**
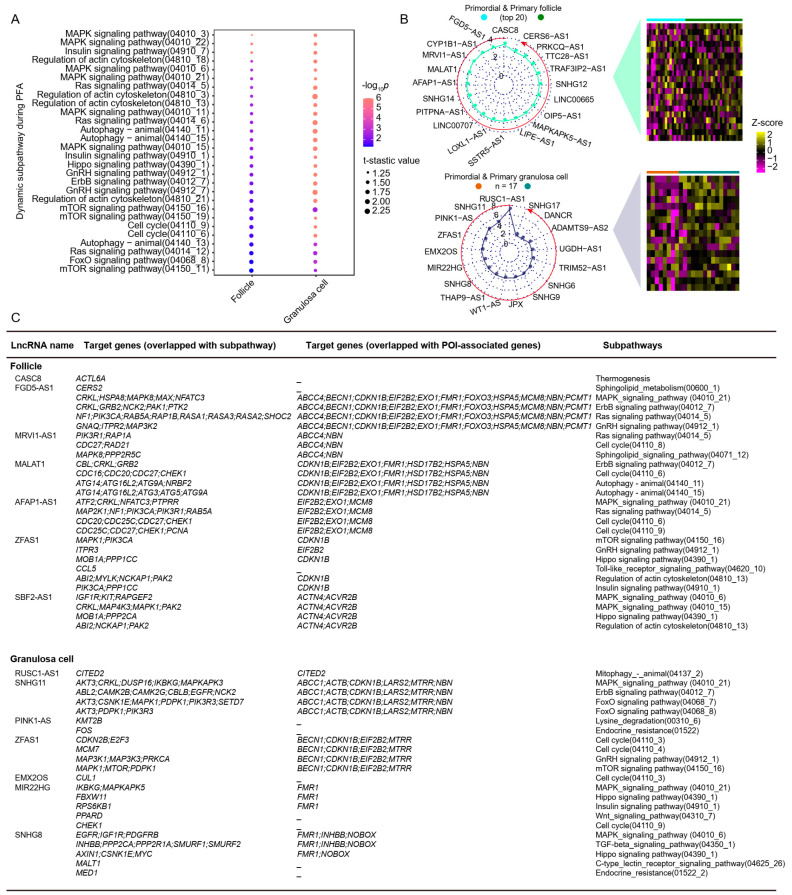
The connection between lncRNAs and subpathways during PFA. (**A**) Subpathway enrichment bubble plot during PFA, a larger *p*-value (−Log_10_) indicates a higher degree of enrichment. (**B**) Radar chart displaying the rank of lncRNAs in follicles (upper panel) and granulosa cells (lower panel) during PFA. The direction of the red circle represents the rank from largest to smallest value. Right heat map showing the expression level of top-rank lncRNAs in follicles and granulosa cells. (**C**) The predicted connection among lncRNAs, target genes, subpathway genes, and POI-related genes.

**Figure 6 ijms-24-10525-f006:**
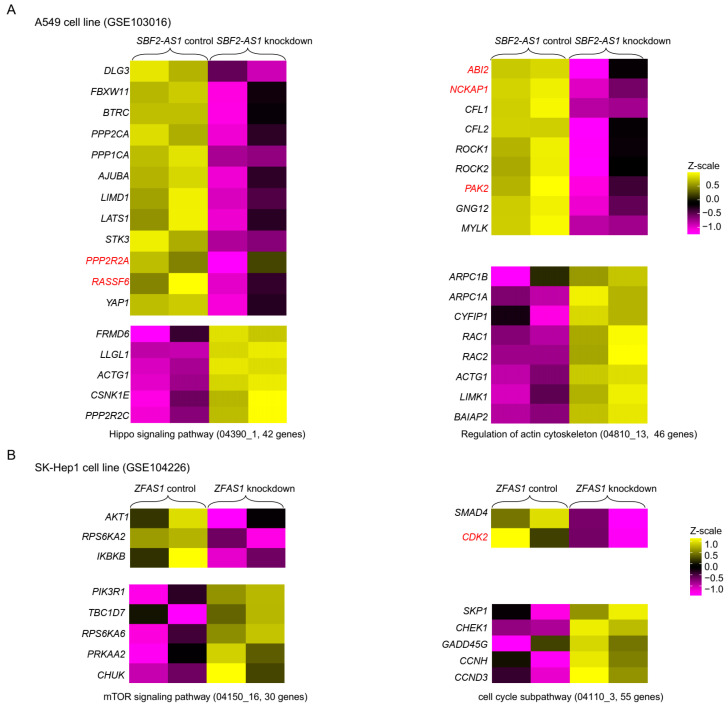
Heat map of genes associated with Hippo, regulation of actin cytoskeleton, mTOR, and cell cycle subpathways (**A**) in control and *SBF2*-*AS1* knockdown A549 cell line (GSE103016) and (**B**) in control and *ZFAS1* knockdown SK-Hep1 cell line (GSE104226). Genes in red are the target gene of lncRNA during PFA.

**Figure 7 ijms-24-10525-f007:**
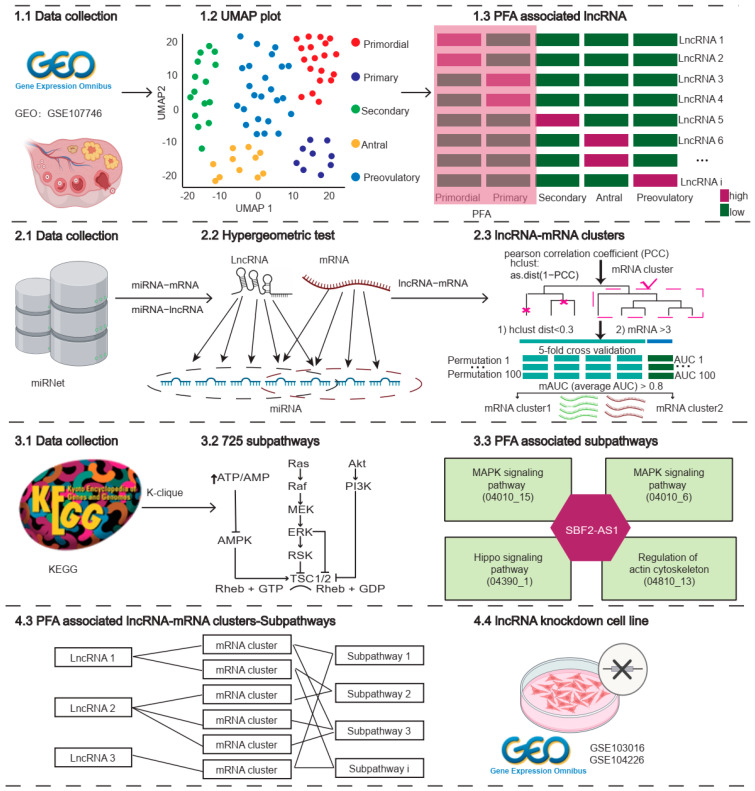
Workflow of identifying PFA-associated lncRNAs. The dark pink color in 1.3 indicates high expression lncRNAs, while the green color indicates low expression lncRNAs.

## Data Availability

The log2-transformed FPKM of lncRNAs used to support this study was obtained in Appendix A (https://www.cell.com/cms/10.1016/j.molcel.2018.10.029/attachment/7d2c901a-6f57-40ce-a135-cd0842d719bc/mmc2.xlsx, accessed on 10 March 2023) of Zhang et al.’s research. The log2-transformed FPKM of genes was downloaded through GEO accession number GSE107746. The control and SBF2-AS1 knockdown A549 cell lines were obtained through GEO accession number GSE103016. The control and ZFAS1 knockdown SK-Hep1 cell lines were obtained through GEO accession number GSE104226.

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
