# Peer review of "A Subpathway and Target Gene Cluster-Based Approach Uncovers lncRNAs Associated with Human Primordial Follicle Activation"

_ijms, 2023, doi:10.3390/ijms241310525_

Round 1

Reviewer 1 Report

In the manuscript, the Authors identified the primordial follicle activation (PFA)- related subpathway and gene cluster to show the activated/inhibited gene modules. The manuscript, including the validation of the results, is entirely based on bioinformatics analyses, therefore from the physiological point of view is hard to follow the specific bioinformatician simulations and in my opinion, Authors should make an effort to confirm the 2-3 typical lncRNA abundance by qPCR. This is a good laboratory practice when the NGS data are laboratory analysed. Nevertheless, below I prepared some major points which may help improve the manuscript's quality.

There is a lack of the hypothesis and aims of the study.

When the Authors described methods - the workflow of the flowchart is necessary for readers who are not familiar with bioinformatic analyses.

In line 53 is mentioned that the GSE107746 was used from Zhang et al. 2018, whereas in line 254 from Pangas et al. It is a mistake, so please carefully check the references. In GEO

data source GSE107746 it is mentioned  6 different datasets for the preovulatory follicles, while the Authors write only n= 3. In my opinion, the Authors must clearly indicate which data they used.

In the text, the Authors should explain why they used all groups i.e.. primordial follicles, primary follicles, secondary follicles, antral follicles, preovulatory follicles, primordial granulosa cells, primary granulosa cells, secondary granulosa cells, antral granulosa cells, preovulatory granulosa cells, when the PFA phenomenon appeared between the primordial and primary follicle stages.

Lines 88-89 – the values of the lncRNA are not consistent with the data in Fig 1 c.

It Is irritating and confusing when the Authors described the groups like follicle and granulosa cells without specification of the maturation stages. Please change it in the whole manuscript, for example, point 4.2 transcriptome data of human ovarian follicles and matched granulosa cells – it meat what… -primary, primordial etc….

There is no simple explanation in the manuscript of when and how Authors define the PFA event.

Lines 276-278- wrong values on protein-coding genes and lncRNA?

Point 4.4.- what for the analysis of miRNA was performed, Authors did not provide any data for this kind of analysis. It should be explained.

Premature ovarian insufficiency (POI)-related gene set – the results of this analysis are mentioned neither in the Results section nor the Discussion, so the question arises why the Authors made this confirmation.

The discussion is of a broad degree of generality. Why Authors did not focus on the potential functions of lncRNA target gene clusters - SBF2-AS1, MALAT1 and ZFAS1 or POI-connected genes (EIF2B2, EXO1, FMR1, FOXO3). Lines 226-227 –‘ lncRNAs that modulated cell proliferation and differentiation have been identified’ – please clearly indicated the examples and give the reference for this assumption.

 Line 232- important functions – which ones (based on the literature)

line 239-241- ‘We noticed that the activation pattern of the subpathway in follicles showed higher heterogeneity than that in granulosa cells.’ - this phenomenon may have physiological consequences?, but what does it mean from the physiological point of view?

There are typos and stylish mistakes.

Reviewer 2 Report

In the current study the authors aimed at asseswsing the role of lncRNAs in the process of follicle maturation. The observed that the expression level of lncRNA was more specific than protein-coding genes in both follicles and granulosa cells and, particularly, the Hippo signaling subpathway was revealed as regulated during maturation.

The study provides a comprehensive description of the lncRNAs expression profile. The manuscript is clear and well written. 

I would suggest the authors to provide the clinical implication of the findings, including the optimization of the in vitro maturation protocols and/or the influence of the controlled ovarian stimulation used in reproductive medicine.
